# Digital Influencers' Attributes and Perceived Characterizations and Their Impact on Purchase Intentions

Manuel Joaquim de Sousa Pereira [1,*] , António Cardoso [2] , Ana Canavarro [3] , Jorge Figueiredo [4] and Jorge Esparteiro Garcia [5]

1   Polytechnic Institute of Viana do Castelo, 4900-347 Viana do Castelo, Portugal
2   Department of Business and Communication Sciences, University Fernando Pessoa, 4294-004 Porto, Portugal; ajcaro@ufp.edu.pt
3   IPAM Lab, IPAM/Europeia University, Rua Manuel Pinto de Azevedo, 748, 4100-320 Porto, Portugal; anacanavarroster@gmail.com
4   Centre for Law, Economic and Environmental Studies (CEJEA), Lusiada University, 4369-006 Porto, Portugal; jorgefig85@gmail.com
5   ADiT-LAB, Instituto Politécnico de Viana do Castelo, Rua Escola Industrial e Comercial Nun'Álvares, 4900-347 Viana do Castelo, Portugal; jorgegarcia@esce.ipvc.pt
*   Correspondence: msousa.manuel@gmail.com

**Abstract:** Research into the role of digital influencers in marketing strategies is a rapidly developing area that has attracted the interest of researchers and organizations. In recent years, organizations have become increasingly interested in using digital influencers to promote their brands and disseminate advertising messages with a high impact on their target audience. Digital influencers are beginning to be used as models for sustainable consumption behavior (for example in the fashion, food, and health sectors) by promoting environmental and sustainable values. By promoting sustainable content and disseminating messages of environmental awareness, digital influencers can help achieve the Sustainable Development Goals (SDGs). This study aims to identify the attributes (attitude homophily, physical attractiveness, and social attractiveness) and perceived characterizations (trustworthiness, perceived expertise, and parasocial relationship) of digital influencers and their impact on purchase intention among a sample of Portuguese consumers. It also aims to identify the most relevant types of digital influencers according to their areas of influence (fashion, sports, beauty, and cinema/TV/music) and their impact on purchase intention. For data collection, an online questionnaire was developed and administered to a non-probabilistic convenience sample. Only respondents who had experience purchasing a product or service after watching a YouTuber's advertisement (screening question) or following or searching for a digital influencer could complete the questionnaire. A total of 243 valid questionnaires were received. The main findings are that the attributes and perceived characterizations of digital influencers have a positive and significant impact on purchase intention. It was also found that digital influencers can enhance shopping experience and credibility, which has a strong impact on consumers' purchase intentions. In terms of sector, the data show that the most important influencer in the 'Fashion' sector is Helena Coelho, in the 'Sports' sector is Cristiano Ronaldo, in the 'Beauty' sector is Sara Sampaio, and in the 'Music, TV, Cinema' sector is Ricardo Araújo Pereira. This study can help companies use digital influencers more effectively in their digital marketing strategies, as credibility, experience, and parasocial relationships have a strong impact on consumers' purchase intention.

**Keywords:** digital influencers; influencer marketing; influencer attributes; perceived characterizations; purchase intention

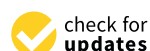



## 1. Introduction

The emergence of social media changed the communication paradigm, as the collaborative and interactive nature of these media enabled ordinary citizens to take an active

role in the communication process. It became possible to create content and participate in conversations within digital communities. In the context of this paradigm shift, the emergence of the digital influencer as an influencer of the purchase decision assumes particular importance in the study of new consumer behaviors, especially those that take place in a digital environment.

The studies identified in the literature [1–4] indicate that the attractiveness and reliability of sustainable digital influencers, as well as the quality and quantity of the content they produce, tend to determine their communicative persuasiveness. The persuasive effectiveness of the digital influencer in the intention to purchase sustainable products depends on the engagement aroused, since the greater the persuasion capacity, the greater the engagement, and, consequently, the greater the intention to purchase sustainable products.

Several studies [1–3] recognize the existence of 'greenfluencers'. This is a type of influencer specialized in promoting a sustainable lifestyle. These influencers are important in educating and raising the awareness of their followers about environmental concerns and sustainable consumption behaviors. The environmental experience of digital influencers has an impact on the intention to purchase green products [5–8].

According to Jaiswal and Kant [9] when buying green products, consumers are simultaneously concerned about the ecological quality of the product and the environmental consequences of their purchasing decision.

The results of the study by Wielki [10] show the growing importance of digital influencers in the functioning of the digital promotion ecosystem and its impact on sustainable development. In the fashion sector, digital influencers can be strategically used by companies and brands to promote the consumption of sustainable clothing, especially in the post-COVID-19 period when online sales among young people are increasing [4].

In order to determine the 'state of the art', a bibliographic search was conducted, consisting of published scientific articles on the following topics: social media marketing; influencer marketing; digital influencers and purchase intent; and neuromarketing tools, advertising effectiveness, and social media.

This research aimed to identify the attributes (attitude, homophily, physical attractiveness, and social attractiveness) and perceived characterizations (trustworthiness, perceived expertise, and parasocial relationship) of digital influencers and their impact on the purchase intention of Portuguese consumers. Specifically, the aim was to (1) identify the most important type of influencer; (2) identify the most relevant digital influencers according to their sector of influence (Fashion, Sports, Beauty, and Cinema/TV/Music); (3) discover the main attributes of influencers; (4) identify the most relevant perceived characterizations of digital influencers; (5) evaluate consumers' purchase intention.

The key research questions for this study were as follows: Do respondents trust digital influencers? Do respondents have a homophilic attitude toward digital influencers? Do respondents find digital influencers physically attractive? Is there a social affinity for digital influencers? Do respondents think digital influencers have experience in their area of influence? Do respondents recognize the digital influencer as an intimate conversation partner (parasocial relationship)? Are respondents willing to buy products promoted by digital influencers (purchase intention)?

In order to achieve the objectives and answer the research questions, a questionnaire was developed using Google Docs and distributed through the authors' social networks, following a convenience sampling process. This process took place from 16 June to 6 November 2022 and consisted of closed-ended questions, evaluated with a five-point Likert-type scale, where 1 means 'I totally disagree' and 5 means 'I totally agree' [11,12].

This article is structured as follows. Section 1 is the introduction. Section 2 is the literature review addressing topics such as social media marketing—concept and importance; influencer marketing and digital influencers; neuromarketing tools, advertising effectiveness, and social media; social media marketing, digital influencers' attributes, perceived characterizations, and purchase intent. Section 3 describes the methodology. Section 4 provides the analysis of the results, and Section 5 is the discussion. Section 6 describes

the theoretical and practical implications. Section 7 discusses the limitations and future research, and Section 8 gives the conclusions.

## 2. Literature Review

### 2.1. Social Media Marketing—Concept and Importance

The emergence of social media platforms over the years has led to a paradigm shift in consumers' online behavior, changing the way they interact with each other and with brands [13].

In particular, the interactive nature of these media has transformed consumers from passive observers of content to active participants who now actually create large amounts of content through their online conversations, interactions, and behaviors [14]. Central to this shift has been the concept of consumer engagement, which recognizes that customers co-create value through these interactions [15].

Prahalad and Ramaswamy [16] note that the change in the role of the consumer from 'isolated' to 'connected' has allowed them to stop being passive and become active in their expressions towards companies. With the emergence of social networks, there has been a real 'revolution in user-generated content, global community and the publication of consumer opinion' [17].

Promotional strategies in these social media are among the main strategies implemented by communication agencies and companies. In fact, in a survey carried out with companies, 98% of them stated that communication in these media is 'very effective' for increasing their presence and economic growth in today's competitive market [18]. In addition, these communication channels allow companies to engage in a timely manner and establish direct contact with their end customers, at a relatively low cost and with high efficiency compared to traditional media [19], making them relevant for companies of all sizes.

A social media marketing (SMM) strategy refers to an organization's set of integrated activities that transform social media communications (networks) and interactions (influences) into useful strategic means to achieve desired marketing outcomes [20]. In addition, relationship marketing has taken on a new dimension with this social media revolution [21].

The interest of organizations in social networks stems from the numerous benefits that organizations can achieve, such as reducing costs, increasing profitability, sharing knowledge, and building commercial partnerships [22]. However, when it comes to knowing whether social networks should be managed by professionals in the field, it is argued that social networks should be seen as a central element in a company's marketing mix and as such should be managed by a professional manager in the field [21].

In the same vein, Tuten and Solomon [23] even stated that a new 'P' has emerged over time and has been added to the traditional marketing mix with the introduction of social networks, resulting in the five 'Ps', namely product, price, promotion, distribution, and participation in social media marketing. They defined the fifth 'P' as the application of social media technologies to create an environment in which the intended value is created to meet stakeholder needs.

### 2.2. Influencer Marketing and Digital Influencers

According to Glucksman [24] it is now possible to find people to represent companies through branded content on social media accounts such as Instagram, Snapchat, Twitter, and YouTube. An effective influencer strategy is no longer an option, but a requirement to compete in today's marketplace [25]. Consumers are looking to other consumers to influence their purchasing decisions.

The rise of social media has given rise to the influencer boom. Influencer marketing has changed the way brands interact with consumers, especially when it comes to lifestyle brands. Today's consumers are distrustful of traditional advertising and can ignore it more easily, making it essential to find new communication strategies that engage consumers [26].

Social media influencers represent a new breed of independent third-party endorsers who shape audience attitudes through blogs, tweets, and the use of other social media channels [27]. Most importantly, consumers choose to follow influencers because they are 'genuine, open and relatable' [25].

In effect, the past decade has seen the emergence of new influencers through the adoption of digital platforms. These individuals, referred to as digital influencers, have linked their public recognition to the production of content on social media [28]. Acting as true advertisers, they promote the goods, services, and ideas of the brands that hire them, thus attempting to persuade individuals to consume the products.

Digital influencers are ordinary users with a high number of followers on their digital channels, and due to this high number of followers, they are able to monetize their audiences by introducing sponsored content in their photos, videos, or texts [26,29].

Broadly speaking, there are three main types of digital influencers [30]: (1) Celebrities: Perform significant functions outside of digital platforms, such as models, actors, athletes, or presenters, and are seen more as a channel for disclosure or endorsement in co-creation with brands; (2) Influencers: Profiles with a larger number of followers present more general topics such as behavior, daily life, fashion, and beauty. They also work on co-creating content with brands; (3) Micro-influencers: Smaller profiles that typically cover niche topics and also co-create content with brands.

In turn, according to Samyroad [31] there are differences between influencers according to their profile (experts, trend-seekers, celebrities, professional YouTubers, and consumers themselves). (1) Experts: These are the profiles that have professional knowledge of what they are talking about. They can be technicians, personal shoppers, professional make-up artists, etc. (2) Celebrities: They are public figures who have a large number of followers and use their social networks to be closer to their fans. (3) Trend-seekers: They are the ones who are always up to date. They are trend hunters who are always the first to find out what is coming and look for it to show their audience. (4) Professional YouTubers: The professional YouTuber is an online personality who posts videos on the YouTube video-sharing platform and usually has their own personal YouTube channel. They develop this activity as a profession, become celebrities, and create their own brand. Many YouTubers also work with brands and sponsors to monetize their content and earn money from their popularity. (5) Consumers: Consumers can also be influencers, thanks to their interest in sharing opinions or recommendations with their followers.

Usually, in the negotiation process with brands, influencers send a press kit with some performance data about their publications on social networks, such as average interaction and reach [30].

To know how digital influencers participate in the production of content, Sette and Pedro [32] found that digital influencers consider the development of creative, innovative, and quality content as a strategic exchange with brands; they are also involved in the production of content during the different stages of message development, in collaboration with these same brands.

In summary [33] more and more brands are investing in influencer marketing, believing that greater selectivity is being awakened by digital influencers, seeking to increase their credibility with their followers.

### 2.2.1. Neuromarketing Tools, Advertising Effectiveness, and Social Media

Neuromarketing is a promising field of study that aims to analyze issues related to advertising effectiveness, product appeal, and, more generally, how to create more effective marketing strategies [34] (Neuroimaging and physiological tools can provide important information about the neural responses of consumers' brains. The study of brain processes and regions involved in consumer purchase intention enriches the development and evaluation of any marketing strategy. Specifically, in advertising research, brain processes such as emotions, feelings, motivation, reward, attention, and memory need to be considered [35].

Social media managers can also use neuromarketing insights to inform their social media strategies. Neuromarketing is becoming increasingly important in predicting user behavior through biometric measurements, so it can be an essential tool for developing content that engages organizations and their audiences [36].

Indeed, it is no secret that for a brand to realize the full marketing potential of social media, it must present a unique voice and personality through which it can communicate and connect with its consumers [37].

Neuromarketing research in social media can assure companies of information on how their communication efforts are perceived by the audience in terms of emotional engagement, memory retention, purchase intention, novelty, awareness, and attention [38].

What about the attention and emotions that influencers can evoke in their followers? Are neuromarketing tools capable of measuring these dimensions? A study [39] carried out in June 2019 in a laboratory at the Complutense University of Madrid, with a sample of 19 girls aged between 11 and 16 concluded that consumers perceive influencers' recommendations as more trustworthy, less commercial, and more natural than traditional digital advertising. The results show that the videos that showcase influencers' content are the ones that generate the most attention peaks.

### 2.2.2. Social Media Marketing, Digital Influencers Attributes, Perceived Characterizations, and Purchase Intent

Balakrishnan et al. [40] investigated the impact of SMM on brand loyalty and purchase intention and found that SMM, especially electronic word of mouth (WOM), online communities, and online advertising, has positive effects on brand loyalty and purchase intention.

In the same sense, Gautam and Sharma [41] emphasized that SMM and customer relationships have a significant and positive impact on customers' purchase intentions.

In turn, Sousa and Alturas [26] found that online media is very important for the 'information research' stage of the purchase decision process, with Facebook, YouTube, and Instagram being the social networks with more users and more digital influencers.

It is important to note that SMM activities and customer experience (CX) can be involved together. These two constructs are interrelated, as all the marketing activities and/or experiences of the company perceived by the customer will influence the customer's response and will be involved in their analysis process prior to the purchase stage [42].

Purchase intention, in turn, is the customer's desire to buy a product [43]. When purchasing products, customers will seek relevant information/knowledge. Once a certain amount of information/knowledge has been gathered, customers tend to analyze, consider, compare, and finally make an effective purchase [42].

Balaban and Szambolics [44] warn that in today's society dominated by digital communication, there is an ongoing discussion about the relevance of authenticity. They argue that social media influencers (SMIs) always strive to remain authentic in their interactions with followers and brands, but that promotional activities pose a challenge to their authenticity. Nevertheless, policy regulations require social media influencers to disclose sponsored content when using a form of native advertising [45].

With regard to the cosmetics industry, Nugroho et al. [46] developed an interesting study that concludes a positive direct influence of social media influencers on the desire to know more about the product. Also, according to this study, attributes such as attractiveness, credibility, expertise, and trustworthiness have a direct influence on the purchase intention of Gen Z.

On the other hand, research on hotel services [47] showed that digital influencers are more persuasive to their audience when they are rated as reliable and physically attractive. The same is true when there is a one-way affective relationship between the endorser and the audience. The public perception of the authenticity of social media influencers (SMIs) is indeed a key driver of their persuasion as brand supporters.

Research by Lee and Eastin [48] suggests that perceived SMI authenticity is a multidimensional construct consisting of sincerity, genuine endorsement, visibility, expertise, and

uniqueness. Each of the five dimensions has varying effects on consumers' evaluation of an SMI, willingness to follow an SMI, and intention to purchase products recommended by SMIs [48].

The number of followers of digital influencers also has a positive impact on source attractiveness and source trustworthiness, according to a study by Kay et al. [45]. This study also found that the number of followers had a positive effect on consumers' purchase intention. When social media influencers have a large number of followers, consumers perceive them as more attractive and trustworthy, which increases purchase intention.

In terms of brand awareness, it was observed that there is a perception of increased awareness through actions with the use of digital influencers. These actions affect the increase in followers on digital platforms and the increase in sales. As a result, the companies involved are interested in continuing this type of partnership [49].

The studies of several [50–54] have identified three constructs and dimensions in the area of influencer endorsement and advertising effectiveness. These are the following: (1) influencer attributes (attitude homophily; physical attractiveness; social attractiveness), (2) perceived characterizations (trustworthiness; perceived expertise; parasocial relationship), and (3) purchase intentions.

In terms of the personal characteristics of influencers, they can be described and explained as follows. Attitudinal homophily refers to the principle that contact between similar people occurs at a higher rate than contact between dissimilar people [55] cited in [54]. Physical attractiveness refers to the impact of appearance in advertising. Traditional studies of physical attractiveness have analyzed the effects of the appearance of the model in advertising [56]. Social attractiveness refers to the likeability of a speaker [51] This attribute is related to the creation of emotions and brand love.

In turn, perceived characterizations (such as trustworthiness; perceived expertise; parasocial relationship) are followers' perceptions based on influencers' attributes. Trustworthiness is related to the credibility of a source or person and the extent to which the source is considered valid [57] cited in Masuda et al. [54]. Perceived expertise and authenticity are usually accompanied by trustworthiness in predicting positive social media marketing outcomes [51,52,54,58,59].

Parasocial relationships (PSRs) are the ongoing relationships that users develop with a mediated performer. Originally studied in the field of traditional mass media communication, in these relationships, the user perceives the performer as an intimate interlocutor [60] with similarities to face-to-face communication. In this sphere, the audience is free to withdraw at any time, but there is involvement and mediated intimacy. According to Masuda et al. [54] PSR has the greatest influence on purchase intention.

Purchase intention refers to the behavioral intentions of influencers' followers [61]. It is known that the perceived characteristics of influencers, generated from their personal attributes, can generate purchase intentions [54].

## 3. Methodology

The aim of the research is to analyze how the attributes and characteristics of digital influencers influence consumers' purchase intentions.

Specifically, the research aims to (1) identify the most important type of influencer; (2) identify the most relevant digital influencers according to their sector of influence (fashion, sports, beauty, and cinema/TV/music); (3) discover the most important attributes of influencers; (4) identify the most relevant characteristics of digital influencers; (5) evaluate consumer purchase intention.

The overall research question was to investigate the role of digital influencers in influencing consumer behavior (purchase intention) among a sample of the Portuguese population. In this context, a number of more specific questions arose that the study sought to answer: Do the respondents trust digital influencers? Do respondents have a homophilic attitude toward digital influencers? Do respondents find digital influencers physically attractive? Is there a social affinity with digital influencers? Do respondents think

digital influencers have experience in their sector of influence? Do respondents recognize the digital influencer as an intimate conversation partner (parasocial relationship)? Are respondents willing to buy products promoted by digital influencers (purchase intention)?

In order to achieve the objectives and answer the research questions, a questionnaire was developed consisting of closed-ended questions rated on a 5-point Likert scale, where 1 means 'strongly disagree' and 5 means 'strongly agree' [11,12].

The questionnaire was based on and adapted from previous studies identified in the literature (Table 1) [51–54,62,63] based on three constructs and dimensions: (1) influencer attributes (attitudinal hominess; physical attractiveness; social attractiveness), (2) perceived characterizations (trustworthiness; perceived expertise; parasocial relationship), and (3) purchase intentions.

**Table 1.** Survey constructs and dimensions.

| Constructs | Dimensions | No. of Items | Author |
|---|---|---|---|
| **Influencer Attributes** | Attitude homophily | 4 | Lou and Kim (2019) [50]<br>Masuda, Han, and Lee (2022) [54] |
| | Physical attractiveness | 4 | Sokolova and Kefi (2020) [51]<br>Masuda, Han, and Lee (2022) [54] |
| | Social attractiveness | 4 | Sokolova and Kefi (2020) [51]<br>Masuda, Han, and Lee (2022) [54] |
| **Perceived Characterizations** | Trustworthiness | 3 | Lou and Kim (2019) [50]<br>Sokolova and Kefi (2020) [51]<br>Schouten et al. (2020) [52]<br>Masuda, Han, and Lee (2022) [54] |
| | Perceived expertise | 4 | Lou and Kim (2019) [50]<br>Sokolova and Kefi (2020) [51]<br>Schouten et al. (2020) [52]<br>Masuda, Han, and Lee (2022) [54] |
| | Parasocial relationship | 8 | Lou and Kim (2019) [50]<br>Sokolova and Kefi (2020) [51]<br>Masuda, Han, and Lee (2022) [54] |
| **Purchase Intentions** | Purchase Intentions | 2 | Casalo et al. (2017) [62]<br>Masuda, Han, and Lee (2022) [54] |

On the other hand, the independent variables include gender, age groups, activity, influencer type, and the main Portuguese digital influencers by area (fashion, sports, beauty, music). The identification of the main Portuguese digital influencers, by each of the 4 areas, took into account the study by Marktest [64].

The questionnaire was created via Google Forms, and after a pre-test [12] with 12 students who did not register the need for adaptation, it was distributed via social media platforms (Facebook, Messenger, Instagram, and WhatsApp) in June and November 2022.

As validated in the study by Masuda, Han, and Lee [54], only respondents who answered yes to three screening questions (regularly use an influencer's YouTube channel; have experience buying products or services after watching a YouTuber's video; subscribe to at least one YouTube channel) or who followed or searched for a digital influence could answer the questionnaire and complete the rest of the questionnaire [61].

The data collected through Google Forms were exported to SPSS 29 for data analysis and statistical treatment. After collecting 286 questionnaires, 243 were validated as 43 did not answer the screening questions.

The respondents are mostly female (64.6%), in the age group 20–29 (33.3%) and <20 years (20.6%), with 135 employees (55.6%) and 108 students (44.4%). The vast majority use digital platforms to shop online (91.2%), mainly 'products and services' (51.9%) (Table 2).

**Table 2.** Sample.

|  |  | F | % |
|---|---|---|---|
| **Gender** | Male | 86 | 35.4 |
|  | Female | 157 | 64.6 |
| **Age Groups** | <20 | 50 | 20.6 |
|  | 20–29 | 81 | 33.3 |
|  | 30–39 | 30 | 12.3 |
|  | 40–49 | 49 | 20.2 |
|  | 50–59 | 22 | 9.1 |
|  | >59 | 11 | 4.5 |
| **Activity** | Worker | 135 | 55.6 |
|  | Student | 108 | 44.4 |
| **Buy Through Digital Platforms** | Only products | 95 | 39.1 |
|  | Only services | 3 | 1.2 |
|  | Products and services | 126 | 51.9 |
|  | Do not buy | 19 | 7.8 |

## 4. Analysis of Results

According to Samyroad [31] there are differences between influencers according to their profile (experts, trend-seekers, celebrities, professional YouTubers, and consumers themselves).

As shown in the following table (Table 3), the most important influencers in online purchase decisions are mainly 'Digital Influencers—Trend Seekers' (35.8%) and 'Experts' (33.3%), followed by 'Celebrities' (9.5%) and 'Professional YouTubers' (8.2%).

**Table 3.** Influencer type.

|  |  | F | % |
|---|---|---|---|
| **Influencer** | Professional YouTuber | 20 | 8.2 |
|  | Celebrity | 23 | 9.6 |
|  | Expert | 81 | 33.3 |
|  | Digital Influencer (trend-seekers) | 87 | 35.8 |
|  | Other | 32 | 13.2 |

In terms of area of influence, we can see (Table 4) that the main influencer in the 'Fashion' field is Helena Coelho (22.2%), in the 'Sports' field is Cristiano Ronaldo (63.8%), in the 'Beauty' field is Sara Sampaio (34.2%), and in the 'Music, TV, Cinema' field is Ricardo Araújo Pereira (34.2%).

**Table 4.** Main digital influencer by area of influence.

| Fashion | F | % | Sports | F | % | Beauty | F | % | Music, TV, Cinema | F | % |
|---|---|---|---|---|---|---|---|---|---|---|---|
| Pipoca mais doce | 18 | 7.4 | Cristiano Ronaldo | 155 | 63.8 | Sara Sampaio | 83 | 34.2 | Rita Pereira | 31 | 12.8 |
| Liliana Filipe | 23 | 9.5 | Pepe | 9 | 3.7 | Vanessa Martins | 10 | 4.1 | Cristina Ferreira | 32 | 13.2 |
| Helena Coelho | 54 | 22.2 | Ricardo Quaresma | 2 | 0.8 | Liliana Filipe | 20 | 8.2 | Daniela Ruah | 26 | 10.7 |
| Vanessa Martins | 9 | 3.7 | Renato Sanches | 6 | 2.5 | Mafalda Sampaio | 10 | 4.1 | Ricardo Pereira | 5 | 2.1 |
| Mafalda Sampaio | 19 | 7.8 | Luis Figo | 12 | 4.9 | Sofia Barbosa | 18 | 7.4 | Ricardo Ar. Pereira | 83 | 34.2 |
| Débora Rosa | 6 | 2.5 | André Gomes | 3 | 1.2 | Inês Rocha | 22 | 9.1 | David Carreira | 7 | 2.9 |
| Other | 114 | 46.9 | Other | 56 | 23.0 | Other | 80 | 32.9 | Other | 59 | 24.3 |

Respondents show 'attitude homophily' (Table 5) toward the preferred digital influencer (M = 3.23 > 2.5; Sd = 1.194), with 47% agreeing with the prepositions presented.

Specifically, respondents agree (57.2%) that 'this digital influencer shares my values' (M = 3.44; Sd = 1.259).

**Table 5.** Attitude homophily (AH).

| Cronbach's Alpha: 0.928 ITEMS | Strongly Disagree 1 | Disagree 2 | Undecided 3 | Agree 4 | Strongly Agree 5 | M | Sd |
|---|---|---|---|---|---|---|---|
| | F (%) | F (%) | F (%) | F (%) | F (%) | | |
| AH1. This digital influencer and I have a lot in common. | 31 (12.8) | 21 (8.6) | 79 (32.5) | 83 (34.2) | 29 (11.9) | 3.24 | 1.168 |
| AH2. This digital influencer and I are a lot alike | 38 (15.6) | 32 (13.2) | 88 (36.2) | 65 (26.7) | 20 (8.2) | 2.99 | 1.166 |
| AH3. This digital influencer thinks like me | 35 (14.4) | 15 (6.2) | 75 (30.9) | 91 (37.4) | 27 (11.1) | 3.25 | 1.184 |
| AH4. This digital influencer shares my values | 33 (13.6) | 15 (6.2) | 56 (23) | 91 (37.4) | 48 (19.8) | 3.44 | 1.259 |
| **TOTAL Mean** | 34 (13.9) | 20 (8.2) | 75 (30.9) | 83 (34.2) | 31 (12.8) | 3.23 | 1.194 |
| **PCA** = Factor 81.4 | **KMO = 0.818/Bartlett test** = 798.071/**Sig** < 0.001 | | | | | | |

The scale shows very good internal consistency (Cronbach's alpha = 0.0928), with correlations between items ranging from 0.696 to 0.828 ($p = 0.000$). The exploratory factor analysis carried out using the principal component analysis (PCA) method revealed the existence of a factor that explains 82.4% of the total variance (KMO = 0.818; $p < 0.001$).

Regarding the dimension 'physical attractiveness' (Table 6), most respondents (43.6%) agree with the prepositions presented (M = 3.43 > 2.5; Sd = 1.288). Specifically, they agree (62.5%) that the preferred digital influencer is 'handsome/pretty' (M = 3.75; Sd = 1.281).

**Table 6.** Physical attractiveness (PA).

| Cronbach's Alpha: 0.839 Items | Strongly Disagree 1 | Disagree 2 | Undecided 3 | Agree 4 | Strongly Agree 5 | M | Sd |
|---|---|---|---|---|---|---|---|
| | F (%) | F (%) | F (%) | F (%) | F (%) | | |
| PA1. I think this digital influencer is handsome/pretty | 25 (10.3) | 10 (4.1) | 56 (23) | 61 (25.1) | 91 (37.4) | 3.75 | 1.281 |
| PA2. This digital influencer is somewhat attractive | 23 (9.5) | 12 (4.9) | 53 (21.8) | 78 (32.1) | 77 (32.1) | 3.72 | 1.229 |
| PA3. I have a better relationship with this digital influencer than with the others | 49 (20.2) | 26 (10.7) | 73 (30) | 61 (25.1) | 34 (14) | 3.02 | 1.316 |
| PA4. I think this digital influencer affects the way I see myself | 42 (17.3) | 19 (7.8) | 65 (26.7) | 71 (29.2) | 46 (18.9) | 3.25 | 1.329 |
| **TOTAL Mean** | 34 (13.9) | 17 (6.9) | 62 (25.6) | 68 (28) | 62 (25.6) | 3.43 | 1.288 |
| **PCA** = Factor 67.5 | **KMO = 0.730/Bartlett test** = 436.020/**Sig** < 0.001 | | | | | | |

Cronbach's alpha analysis shows good internal consistency (α = 0.839), with correlations between items ranging from 0.443 to 0.756 and being significant ($p = 0.000$). The exploratory factor analysis carried out using the principal component analysis (PCA) method revealed the existence of a factor that explains 67.5% of the total variance (KMO = 0.730; $p < 0.001$).

The 'social attractiveness' (Table 7) of the digital influencer is rated positively by respondents (M = 3.22 > 2.5; Sd = 1.335), with 38.6% agreeing with the statements presented. The scale shows good internal consistency (α = 0.891) and the correlations between the items vary between 0.539 and 0.766 ($p = 0.000$). The exploratory factor analysis carried out using the principal component analysis (PCA) method revealed the existence of a factor that explains 75.4% of the total variance (KMO = 0.771; $p < 0.001$).

**Table 7.** Social attractiveness (SA).

| Cronbach's Alpha: 0.891 Items | Strongly Disagree 1 | Disagree 2 | Undecided 3 | Agree 4 | Strongly Agree 5 | M | Sd |
|---|---|---|---|---|---|---|---|
| | F (%) | F (%) | F (%) | F (%) | F (%) | | |
| SA1. I think this digital influencer could be my friend | 50 (20.6) | 29 (11.9) | 78 (32.1) | 57 (23.5) | 29 (11.9) | 2.94 | 1.288 |
| SA2. I want to have a friendly chat with this digital influencer | 41 (16.9) | 19 (7.8) | 54 (22.2) | 67 (27.6) | 62 (25.5) | 3.37 | 1.386 |
| SA3. We could be able to establish a personal friendship with each other | 52 (21.4) | 23 (9.5) | 81 (33.3) | 49 (20.2) | 38 (15.6) | 2.99 | 1.336 |
| SA4. This digital influencer would be pleasant to be with | 31 (12.8) | 7 (2.9) | 35 (14.4) | 75 (30.9) | 95 (39.1) | 3.81 | 1.330 |
| **TOTAL Mean** | 43 (17.7) | 20 (8.2) | 62 (25.5) | 62 (15.5) | 56 (23.1) | 3.22 | 1.335 |
| **PCA** = Factor 75.4 | **KMO = 0.771/Bartlett test** = 617.259/**Sig** < 0.001 | | | | | | |

In general, as shown in the following table (Table 8), respondents seem to trust their favorite digital influencer (M = 4.01; Sd = 1.183), with 77% agreeing with the statements presented. Cronbach's alpha analysis shows good internal consistency ($\alpha$ = 0.968), with high (0.901 and 0.914) and significant ($p$ = 0.000) correlations between items. Principal component analysis (PCA) revealed the existence of a factor explaining 94.1% of the total variance (KMO = 0.784; $p$ < 0.001).

**Table 8.** Trustworthiness (Trust).

| Cronbach's Alpha: 0.968 Items | Strongly Disagree 1 | Disagree 2 | Undecided 3 | Agree 4 | Strongly Agree 5 | M | Sd |
|---|---|---|---|---|---|---|---|
| | F (%) | F (%) | F (%) | F (%) | F (%) | | |
| Trust1. I feel this digital influencer is honest | 20 (8.2) | 5 (2.1) | 32 (13.2) | 81 (33.3) | 105 (43.2) | 4.01 | 1.180 |
| Trust2. I consider this digital influencer trustworthy | 20 (8.2) | 7 (2.9) | 27 (11.1) | 89 (36.6) | 100 (41.2) | 4.00 | 1.177 |
| Trust3. I feel this digital influencer truthful | 20 (8.2) | 5 (2.1) | 32 (13.2) | 79 (32.5) | 107 (44) | 4.02 | 1.183 |
| **TOTAL Mean** | 20 (8.2) | 6 (2.4) | 30 (12.4) | 83 (34.2) | 104 (42.8) | 4.01 | 1.18 |
| **PCA** = Factor 94.1 | **KMO = 0.784/Bartlett test** = 908.923/**Sig** < 0.001 | | | | | | |

Respondents consider their digital influencer to be an 'expert' (Table 9) in their area of activity and influence (M = 3.89 > 2.5; Sd = 1.178), with the majority (73.5%) agreeing with the statements made. In particular, they agree (77.4%) that the preferred digital influencer is experienced enough to make judgments about their area of expertise (M = 4.00; Sd = 1.189). The scale has good internal consistency ($\alpha$ = 0.944) with high (0.765 and 0.907) and significant ($p$ = 0.000) correlations between items. Principal component analysis (PCA) revealed the existence of a factor that explains 85.7% of the total variance (KMO = 0.836; $p$ < 0.001).

**Table 9.** Expertise (Exp).

| Cronbach's Alpha: 0.944 Items | Strongly Disagree 1 | Disagree 2 | Undecided 3 | Agree 4 | Strongly Agree 5 | M | Sd |
|---|---|---|---|---|---|---|---|
| | F (%) | F (%) | F (%) | F (%) | F (%) | | |
| Exp1. I think this digital influencer knows a lot | 22 (9.1) | 9 (3.7) | 41 (16.9) | 108 (44.4) | 63 (25.9) | 3.74 | 1.154 |
| Exp2. This digital influencer is competent to make assertions about things that this digital influencer is good at | 22 (9.1) | 6 (2.5) | 35 (14.4) | 90 (37) | 90 (37) | 3.91 | 1.194 |
| Exp3. I consider this influencer an expert on his/her are | 20 (8.2) | 7 (2.9) | 39 (16) | 87 (35.8) | 90 (37) | 3.91 | 1.176 |
| Exp4. I consider this influencer sufficiently experienced to make assertions about his/her area | 20 (8.2) | 8 (3.3) | 27 (11.1) | 85 (35) | 103 (42.4) | 4.00 | 1.189 |
| **TOTAL Mean** | 21 (8.6) | 7 (3.1) | 36 (14.6) | 93 (38.1) | 86 (35.6) | 3.89 | 1.178 |
| **PCA** = Factor 94.1 | **KMO = 0.836/Bartlett test** = 965.187/**Sig** < 0.001 | | | | | | |

The study shows that the respondents evaluate the dimension 'parasocial relationship' positively (M = 3.5 > 2.5; Sd = 1.334), as 58.6% agree with the statements presented (Table 10). Specifically, respondents agree (72%) that 'this influencer makes me feel comfortable' (M = 3.89; Sd = 1.230) and agree (67.5%) that 'I try to follow this influencer on the social networks/channels and means in which he shares digital content' (M = 3.74; Sd = 1.359). The scale has good internal consistency ($\alpha$ = 0.939) in the sense that the correlations between the items vary between 0.442 and 0.845 and are significant (*p* = 0.000). Principal component analysis (PCA) revealed the existence of a factor that explains 70.9% of the total variance (KMO = 0.836; *p* < 0.001).

**Table 10.** Parasocial relationship (PSR).

| Cronbach's Alpha: 0.939 Items | Strongly Disagree 1 | Disagree 2 | Undecided 3 | Agree 4 | Strongly Agree 5 | M | Sd |
|---|---|---|---|---|---|---|---|
| | F (%) | F (%) | F (%) | F (%) | F (%) | | |
| PSR1. This influencer makes me feel comfortable | 24 (9.9) | 5 (2.1) | 39 (16) | 81 (33.3) | 94 (38.7) | 3.89 | 1.230 |
| PSR2. I try to follow this influencer in the next publications | 31 (12.8) | 15 (6.2) | 35 (14.4) | 77 (31.7) | 85 (35) | 3.70 | 1.344 |
| PSR3. I try to follow this influencer on the social networks/channels and means by which he/she shares digital content | 30 (12.3) | 17 (7) | 32 (13.2) | 71 (29.2) | 93 (38.3) | 3.74 | 1.359 |
| PSR4. This influencer seems to understand the kind of things I want to know | 34 (14) | 22 (9.1) | 69 (28.4) | 74 (30.5) | 44 (18.1) | 3.30 | 1.264 |
| PSR5. If I saw a story about this influencer in a newspaper or magazine, I would read it | 54 (22.2) | 31 (12.8) | 63 (25.9) | 60 (24.7) | 35 (14.4) | 2.96 | 1.359 |
| PSR6. I constantly follow this influencer's posts | 36 (14.8) | 28 (11.5) | 43 (17.7) | 79 (32.5) | 57 (23.5) | 3.38 | 1.354 |
| PSR7. I try to follow this influencer's publications on social networks and YouTube | 40 (16.5) | 23 (9.5) | 42 (17.3) | 73 (30) | 65 (26.7) | 3.41 | 1.401 |
| PSR8. I would like to meet this influencer in person | 30 (12.3) | 18 (7.4) | 42 (17.3) | 62 (25.5) | 91 (37.4) | 3.68 | 1.365 |
| **TOTAL Mean** | 35 (14.4) | 20 (8.2) | 45 (18.8) | 72 (29.6) | 71 (29) | 3.50 | 1.334 |
| **PCA** = Factor 94.1 | **KMO = 0.918/Bartlett test** = 1705.222/**Sig** < 0.001 | | | | | | |

Most respondents (43.2%) agree that they will buy products or services recommended by their favorite digital influencer in the future (M = 3.14; Sd = 1.251). Thus, as can be seen in the following table (Table 11), the data show that respondents 'will buy products or services recommended by this digital influencer' (M = 3.16; Sd = 1.219) and 'are likely to buy products or services after watching this digital influencer' (M = 3.12; Sd = 1.283). The purchase intention scale has good internal consistency ($\alpha$ = 0.922). The correlation between the items is high (0.856) and significant ($p$ = 0.000). Principal component analysis (PCA) revealed the existence of a factor that explains 92.8% of the total variance (KMO = 0.500; $p < 0.001$).

**Table 11.** Purchase intention (PI).

| Cronbach's Alpha: 0.922 Items | Strongly Disagree 1 | Disagree 2 | Undecided 3 | Agree 4 | Strongly Agree 5 | M | SD |
|---|---|---|---|---|---|---|---|
| | F (%) | F (%) | F (%) | F (%) | F (%) | | |
| PI1. I think I will buy products or services recommended by this digital influencer | 37 (15.2) | 23 (9.5) | 75 (30.9) | 79 (32.5) | 29 (11.9) | 3.16 | 1.219 |
| PI2. I will probably buy products or services after watching this digital influencer | 43 (17.7) | 22 (9.1) | 75 (30.9) | 68 (28) | 35 (14.4) | 3.12 | 1.283 |
| **TOTAL Mean** | 40 (16.4) | 23 (9.4) | 75 (30.9) | 73 (30.1) | 32 (13.2) | 3.14 | 1.251 |
| **PCA** = Factor 94.1 | **KMO = 0.500/Bartlett test** = 318.122/**Sig** < 0.001 | | | | | | |

All correlations (Table 12) between the variables are positive and significant ($p < 0.01$). In particular, the correlation between 'trustworthiness' and 'competence' (r = 0.839) stands out. The data also show strong correlations between attitude homophily and parasocial relationships (r = 0.781), between expertise and parasocial relationships (r = 0.781), between trustworthiness and parasocial relationships (r = 0.779), between physical attractiveness and parasocial relationships (r = 0.778), and between social attractiveness and parasocial relationships (r = 0.757). The data also show that all dimensions are positively and significantly correlated with 'Purchase Intentions': PSR (r = 0.713), PA (r = 0.640), AH (r = 0.626), Trust (r = 0.596), Exp (r = 0.578), SA (r = 0.565).

**Table 12.** Correlations.

| | M | SD | (AH) | (PA) | (SA) | (Trust) | (Exp) | (PSR) | (PI) |
|---|---|---|---|---|---|---|---|---|---|
| Attitude Homophily **(AH)** | 3.23 | 1.194 | - | **0.710 **\*\*** | 0.699 \*\* | 0.692 \*\* | 0.654 \*\* | **0.781 \*\*** | 0.626 \*\* |
| Physical attractiveness **(PA)** | 3.43 | 1.288 | | - | 0.664 \*\* | 0.669 \*\* | 0.686 \*\* | **0.778 \*\*** | 0.640 \*\* |
| Social attractiveness **(SA)** | 3.22 | 1.335 | | | - | 0.657 \*\* | 0.648 \*\* | **0.757 \*\*** | 0.565 \*\* |
| Trustworthiness (Trust) | 4.01 | 1.18 | | | | - | **0.839 \*\*** | **0.779 \*\*** | 0.596 \*\* |
| Expertise **(Exp)** | 3.89 | 1.178 | | | | | - | **0.781 \*\*** | 0.578 \*\* |
| Parasocial relationship **(PSR)** | 3.50 | 1.334 | | | | | | - | **0.713 \*\*** |
| Purchase intention **(PI)** | 3.14 | 1.251 | | | | | | | - |

\*\* The correlation is significant at the 0.01 level (2-tailed).

## 5. Discussion

This study was able to identify the relationship between the attributes and perceived characteristics of digital influencers and their impact on purchase intent. Likewise, it was possible to identify the type of influencer by area of influence (fashion, sports, beauty, and cinema/TV/music).

As identified in previous studies [51–54,62,63] this study revealed that respondents have a 'homophily attitude' and trust digital influencers who are considered experts in their area of influence. Similarly, they positively evaluate the physical characteristics and social attractiveness of digital influencers, and it is evident that they positively evaluate the 'parasocial relationship'.

Although the literature does not make a clear distinction between the personal characteristics of influencers and the characterizations of followers, the study has adopted the distinction made in the study by Masuda, Han, and Lee [54], as mentioned above.

Thus, and in the same sense as the evidence identified in the literature [50–54], in terms of the personal attributes of digital influencers, the study revealed the existence of 'attitude homophily' regarding the preferred digital influencer. Similarly, 'physical attractiveness' and 'social attractiveness' are attributes of digital influencers that are positively rated by respondents.

Regarding the 'perceived characteristics' of digital influencers, similar to other studies [50–52,54] it was possible to verify that respondents trust their digital influencers and recognize their perceived expertise, considering them as intimate conversation partners (parasocial relationship—PSR).

Most respondents believe they will purchase products or services recommended by their favorite digital influencer in the future. The results show that the purchase intention induced by an influencer is influenced by 'personal attributes' (attitude homophily (AH), physical attractiveness (PA) and social attractiveness (SA), 'perceived characteristics' (trustworthiness, perceived expertise, parasocial relationship), which confirms the conclusions of the research by Masuda, Han, and Lee [54] which presents trustworthiness as a determinant of followers' purchase intention. They also suggest that PSR may play a more important role in influencing followers' behavioral intentions.

The results of this study are consistent with the findings of Lee et al. [25] and Nugroho et al. [46]. According to their studies, parasocial interaction (PSI) positively influences the intention to purchase a recommended fashion product.

This result is in line with the findings of Gomes, Marques, and Dias [61], who showed that the attributes of the digital influencer and the quality of the content can positively influence the intention to purchase the products recommended by the influencer.

They also support the research of Adrielly and Marconi Freitas [47], who showed that digital influencers are more persuasive to their audience when they are perceived as trustworthy and physically attractive. According to Lee and Eastin [25] influencer authenticity (sincerity, genuine endorsements, visibility, experience, and uniqueness) has varying effects on consumers' evaluation, willingness to follow the influencer, and intention to purchase the products and services they recommend.

However, it was not possible to analyze whether there are significant effects of gender or age, as well as the psychological state of the 'digital follower' respondents in the composition of the PSR. These issues could be explored in future studies.

## 6. Theoretical and Practical Implications

Previous research on digital influencers in Portugal is scarce, as the topic is new. The international studies identified in the literature show the importance and interest of the topic for both academics and organizations, but recognize the need to develop and deepen research in this area of knowledge.

Based on this research need, this study sought to analyze and understand the identification of perceived attributes and characteristics of digital influencers and their impact on purchase intention among a sample of Portuguese consumers. It also aimed to identify the most relevant types of Portuguese digital influencers according to their areas of influence (fashion, sports, beauty, and cinema/TV/music) and their impact on purchase intention.

The data obtained are consistent with the studies identified in the literature, namely the works of [51,54,62,63], as it was found that purchase intention is correlated with

the attributes of the digital influencer and their characteristics. In particular, 'perceived experience' and 'parasocial relationship' appear to influence purchase intention.

Considering the influence of attributes and perceived characteristics of digital influencers on purchase intention, information can be provided to companies and advertising agencies to identify the most relevant influencers, taking into account their target audience and area of influence. This study can help companies use digital influencers more effectively in their digital marketing strategies, as credibility, experience, and parasocial relationship have a strong impact on consumer purchase intention.

This research can be used by companies and influencers to understand how consumers feel about digital influencers, what makes a campaign effective, and why some collaborations fail.

### 7. Limitations and Future Research

The findings suggest that digital influencers can be crucial in the effective development of marketing campaigns with commercial companies and social organizations (e.g., promoting social causes), given the relationships they have with their followers and the trust consumers have in them.

Based on the insights, findings, and limitations of this research, several lines of future research can be explored. This study used a small sample (nonprobability convenience sample), which makes it impossible to generalize the conclusions. This research is limited to the responses of the Portuguese sample and the scales used in this study. In the future, it is recommended that the questionnaire be applied to a larger sample, with different age groups and other geographical areas (comparing regions, countries, genders, etc.). It would also be interesting for future research to evaluate the impact of neuromarketing tools on advertising effectiveness and social media.

Collaborations between digital influencers and organizations from different sectors (profit and non-profit) to produce videos and content that communicate specific messages from brands and social causes can be promising. YouTubers' videos, narratives, and other content can be complemented by posts on other social media platforms such as Facebook, Instagram, Snapchat, Twitter, and TikTok.

Another interesting area, based on the Sustainable Development Goals (SDGs), would be to evaluate the use of digital influencers in promoting health and adopting healthier and more sustainable consumption habits. Similarly, it would be interesting to create new areas for future research, such as how digital influencers can develop creative skills and innovative solutions to continue conquering the public.

In terms of data analysis, it will be opportune in the future to expand the sample and use structural equation models to determine the impact between dimensions.

### 8. Conclusions

The advent of social media has changed the communication paradigm. Consumers are no longer passive and isolated observers but are constantly connected and active in their interactions with companies and the world at large. Through social networks, a true revolution has taken place through the creation of content produced by consumers themselves and their organization into digital communities.

In the context of social media and today's consumer distrust of traditional advertising, the figure of the digital influencer is emerging, who acts as a real advertiser, spreading the goods, services, and ideas of the brands that hire them.

The literature shows [1,2,5–9] that the persuasive ability of sustainable digital influencers depends on the personal characteristics of the influencers (for example, their attractiveness) and the content they produce. Trust in the digital influencer generates greater engagement, having a positive and significant impact on sustainable product intentions.

Influencers' ability to persuade their audiences increases when they are perceived as trustworthy and physically attractive.

The public's perception of the authenticity of social media influencers (SMIs)—a multidimensional construct consisting of sincerity, genuine endorsement, visibility, expertise, and uniqueness—is one of the main drivers of their persuasiveness as brand advocates, with varying effects on the evaluation, followership, and intention to purchase the products they recommend. In this sense, the most important influencers in online purchase decisions are mainly the 'Digital Influencers' and the 'Expert'.

In terms of areas of intervention, the main influencer in the Fashion sector is Helena Coelho; in the Sports sector, it is Cristiano Ronaldo; in the Beauty sector, it is Sara Sampaio; and in the Music, TV, and Cinema sector, it is Ricardo Araújo Pereira.

The results of the study show that the purchase intention induced by the digital influencer is influenced by the 'attributes of the influencer' and by the 'perceived characteristics'. Specifically, it is possible to verify that purchase intention is influenced by 'attitude homophily', the 'physical and social attractiveness' of the influencer, 'trustworthiness', 'perceived expertise', and 'parasocial relationship'. It is this last dimension (parasocial relationship) that has the strongest influence on purchase intention ($r = 0.713$).

On the other hand, parasocial relationship (PSR) is positively and significantly correlated with all other dimensions ($r > 0.757$).

Also noteworthy is the correlation that exists between the perceived expertise dimension and trustworthiness ($r = 0.839$), showing that recognition of the digital influencer's experience influences trust in the influencer.

Further research is needed to understand consumer perceptions of the use of digital influencers for advertising purposes.

**Author Contributions:** We declare that the authors, M.J.d.S.P. and A.C. (António Cardoso), contributed to the conceptualization; methodology and software, A.C. (António Cardoso); Validation, M.J.d.S.P. and A.C. (António Cardoso); Formal analysis and investigation, A.C. (Ana Canavarro); Resources and Visualization, J.F., Supervision and Administration, J.E.G. All authors have read and agreed to the published version of the manuscript.

**Funding:** This research received no external funding.

**Institutional Review Board Statement:** Not applicable.

**Informed Consent Statement:** Informed consent was obtained from all individuals involved in the study.

**Data Availability Statement:** The data that support this research can be seen at this link: https://forms.gle/tTegyAkEpwPjJMvr6 (accessed on 20 June 2023).

**Conflicts of Interest:** The authors declare no conflict of interest.

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
