# Peer review of "Digital Influencers’ Attributes and Perceived Characterizations and Their Impact on Purchase Intentions"

_sustainability, doi:10.3390/su151712750_

Round 1
Reviewer 1 Report
REVIEW COMMENTS
I have only a few concerns about the paper and some suggestions that maybe the authors could consider:
1. Initially, it is important to address the presence of typographical errors and grammatical inaccuracies within the text. Furthermore, the inclusion of lengthy sentences may result in reader confusion and should be carefully considered for improved clarity and comprehension.
2. The authors should explain why they chose Google Scholar, ABI/Inform, ProQuest, and Science d\Direct database to extract data rather than Scopus and WOS database besides Google Scholar.
3. The authors should explain why they chose 5-point Likert scales rather than 7 or 9 -point Likert scales.
4. In the 'Introduction' section, the proposed research gap and the stated objectives do not meet the criteria of proper synergy. Please make the research gap and the research objectives consistent with each other.
5. I think the “Literature Review” can be developed. I suggest some references "neuromarketing tools used in the marketing mix: a systematic literature and future research agenda", "consumer behaviour to be considered in advertising: a systematic analysis and future agenda", and "biomedical technology in studying consumers’ subconscious behavior", which can be useful for improving the section.
6. The authors should write the organizational structure of the paper at the end of the “Introduction” section to ease readers to know the content of the paper.
7. The authors said that new “P” has been added to marketing-mix ”Ps”, which refer to “participation or social media marketing”, the authors should explain this because social media marketing is already under promotion section, in digital or online marketing section. In the same, the authors said “Ps” abbreviation and write distribution, I suggest the authors to write place rather than distribution to correspond with “Ps”.
8. The first paragraph under “influencer marketing and digital influencers” should be written in a more comprehensive way.
9. Again, in the same section, “More…decisions” then “The…brands”. The authors should be more careful and professional when are writing paragraphs. These are not paragraphs. Paragraph content at least 7-8 sentences or 2-3sentences. I suggest the authors rewrite this section in a more professional way.
10. The authors have written “N of itens”, what is that “itens”? Is the authors mean “items”. The authors should be more careful.
11. What criteria does the authors base on to select the experimental sample (number of questionnaires sent out and received)? As stated in the article, the authors have obtained a valid piece of 243. What is the statistical significance of 243 questionnaires for this study?
12. The author(s) should illustrate how many copies they sent and how many copies received?
13. The authors should explicitly state the novel contribution of this work and its similarities and differences with their previous publications.
14. The authors need to clearly articulate the academic as well as practical implications of this study in a separate section which can be named the theoretical and practical implications of this study.
15. The authors need to clearly articulate the limitations and future research of this study in a separate section which can be named ‘limitations and future research’ behind the conclusion section.
16. For readers to quickly catch your contributions, it would be better to highlight major difficulties and challenges and your original achievements to overcome them in a clearer way in the abstract and introduction.
17. What are the challenges of using this tool to examine the facial expressions of consumers?
18. How could/should your study help future studies?
If these revisions can be made to the manuscript, I believe that this study can be accepted for publication.
I wish the authors all the very best with this study.
Initially, it is important to address the presence of typographical errors and grammatical inaccuracies within the text. Furthermore, the inclusion of lengthy sentences may result in reader confusion and should be carefully considered for improved clarity and comprehension.
Author Response
Dear reviewers, Attached is the article with the changes and observations suggested by the reviewers. Grateful for the attention. Manuel Sousa Pereira

Reviewer 2 Report
The article presents an interesting study on social media marketing and the impact of digital influencers on consumers’ purchasing decisions. The paper is generally well structured and rather meets the standards of scientific papers. The literature review includes up-to-dated publications, and the statistical analysis and results are clearly presented. All the comments included in the attached document are meant to improve the quality of the presented study. I do hope you find them useful.

The paper needs proofreading
Author Response
Dear reviewer, Submission attached A grid with changes and observations suggested by reviewers. Grateful for the attention. Manuel Sousa PereiraRound 2
Reviewer 1 Report
REVIEW COMMENTS
I have only a few concerns about the paper and some suggestions that maybe the authors could consider:
1. Initially, it is important to address the presence of typographical errors and grammatical inaccuracies within the text. Furthermore, the inclusion of lengthy sentences may result in reader confusion and should be carefully considered for improved clarity and comprehension.
2. I think the “Neuromarketing tools, Advertising Effectiveness and Social media” can be developed. I suggest some references " exploring global trends and future directions in advertising research: a focus on consumer behavior ", "biomedical technology in studying consumers’ subconscious behavior", and “neuromarketing research in the last five years: a bibliometric analysis”, which can be useful for improving the section.
3. The authors should explicitly state the novel contribution of this work and its similarities and differences with their previous publications.
4. The authors need to clearly articulate the academic as well as practical implications of this study in a separate section which can be named the theoretical and practical implications of this study.
5. The authors need to clearly articulate the limitations and future research of this study in a separate section which can be named ‘limitations and future research’ behind the conclusion section.
6. For readers to quickly catch your contributions, it would be better to highlight major difficulties and challenges and your original achievements to overcome them in a clearer way in the abstract and introduction.
If these revisions can be made to the manuscript, I believe that this study can be accepted for publication.
I wish the authors all the very best with this study.
Initially, it is important to address the presence of typographical errors and grammatical inaccuracies within the text. Furthermore, the inclusion of lengthy sentences may result in reader confusion and should be carefully considered for improved clarity and comprehension.
Author Response
Dear reviewers, We have posted a point-by-point grid of responses to reviewers' comments as requested. We also made a revision, in English, to improve the entire text. grateful for the attention. Best regards, Manuel Sousa Pereira
Reviewer 2 Report
I have read the improved paper carefully and must admit that the changes introduced by the Authors increased its quality. There are a few suggestions that the Authors decided to omit, which is fine, but there are other suggestions I would insist on incorporating them into the article (see the attached document). There is a novum in the paper concerning neuromarketing. It is an interesting topic, but much wider than social media marketing or digital influencer marketing. As there is no relationship with the empirical part of the paper, it is superfluous and thus brings no value to the article.

Author Response

(The authors gave the same response as above.)
